# Elevated cardiovascular risk factors and chronic disease mortality in the Caribbean: A Cross-Sectional Study

Cesar Barrabi[1]*, Celia Foster[2]

**1** Illinois College of Osteopathic Medicine, The Chicago School, Chicago, Illinois, United States of America, **2** Department of Medical Education, Western Atlantic University School of Medicine, Freeport, Grand Bahama Island, The Bahamas

* cbarrabi@thechicagoschool.edu

## Abstract

Cardiovascular disease (CVD) remains a leading cause of premature mortality in the Caribbean, with particularly high rates from stroke and hypertensive heart disease. While global outcomes have improved, CVD-related mortality in the Caribbean remains elevated. This study examined sex-specific CVD mortality, risk factors, and health policy indicators across non-Latin Caribbean nations, using data from the 2019 PAHO Enlace Portal and comparing results to North America (United States and Canada). Despite North Americans exhibiting more overall risky CVD behaviors, CVD mortality was higher in the Non-Latin Caribbean compared to North America (196.7 vs. 122.6 deaths per 100,000). Total-to-HDL cholesterol ratios were significantly higher in the Caribbean, both in men (4.01 vs. 3.63; p < 0.0001) and women (3.82 vs. 3.01; p < 0.0001), indicating reduced cardioprotective effects of HDL. Hypertensive heart disease mortality was significantly higher in Caribbean women compared to North American women (p < .01), with a similar but non-significant trend in men. Caribbean men also had significantly higher rates of undiagnosed hypertension compared to North American men (47.6% vs. 20.8%; p < .001). NCD policy implementation across the Caribbean was inconsistent, with major gaps in CVD drug therapy access, alcohol advertising restrictions, NCD planning, and surveillance systems. These findings underscore the urgent need for regional investment in prevention, policy enforcement, and culturally relevant strategies to reduce CVD disparities and improve health outcomes in the Caribbean.

## Introduction

Cardiovascular disease (CVD) is a leading cause of premature mortality in the Caribbean, with stroke and hypertensive heart disease contributing disproportionately to the region's overall burden [1,2]. Yet unlike other regions in the Americas, the Caribbean has seen little progress in reducing CVD mortality. In previous studies, women

**Data availability statement:** All data used in this study are publicly available through the Pan American Health Organization's (PAHO) ENLACE data portal, accessible at: https://www.paho.org/en/enlace.

**Funding:** The authors received no specific funding for this work.

**Competing interests:** The authors have declared that no competing interests exist.

were found to face elevated rates of hypertensive heart disease, stroke-related deaths, and years of life lost [3]. While several studies have examined CVD trends across Latin America and the Caribbean, most aggregate the data or omit sex-specific insights, limiting their usefulness for targeted intervention [4]. This challenge is compounded by limited funding, poor research infrastructure, and minimal collaboration across Caribbean nations, all of which continue to restrict regional reporting, surveillance, and investment in CVD research [5].

High-income countries such as the United States and Canada have made substantial gains in reducing CVD mortality through decades of investment in prevention, early detection, and health systems strengthening [4,6]. Strong surveillance systems, integrated primary care, population-wide interventions, and clear clinical guidelines have all helped reduce deaths from ischemic heart disease and stroke. However, even these nations continue to face challenges, including significant disparities in healthcare access among rural and Indigenous populations underscoring that progress has not been universally achieved [6]. Additionally, demographic projections suggest CVD burdens in the United States will continue to rise due to an aging population and increased obesity, potentially overwhelming healthcare resources and exacerbating existing inequalities [7].

Yet even within these high-income settings, disparities persist. Racialized populations remain underdiagnosed, underrepresented in clinical trials, and less likely to receive evidence-based treatment [8,9]. Differences in symptom presentation and biological risk, along with provider bias and systemic barriers, contribute to poorer outcomes in these groups. These factors limit access to care and affect the quality of treatment. These inequities are further compounded by social determinants of health, including systemic racism, economic marginalization, and unequal access to care—factors that are even more pronounced in the Caribbean context.

In this study, we analyzed cardiovascular mortality, behavioral and clinical risk factors, and NCD policy indicators across 16 countries in the Americas, with a specific focus on the Non-Latin Caribbean. Our objectives were to compare regional and sex-specific differences in cardiovascular mortality between the Non-Latin Caribbean and North America, to assess disparities in risk factor profiles contributing to CVD, and to evaluate the extent of national-level NCD policy implementation related to cardiovascular prevention and management. This regionally disaggregated, population-level approach aimed to identify key disparities and inform future health system responses across the Caribbean context.

## Methods

### Ethical considerations

This analysis relied solely on publicly accessible, aggregated data from international agencies and did not involve any individual-level or human subjects research. As such, ethical approval was not necessary for this study.

### Study design and data collection

This was a cross-sectional study that examined sex-specific CVD outcomes across 16 Caribbean countries. All data were obtained from the Pan American Health

Organization's (PAHO) Enlace Data Portal, which compiles standardized national health statistics across the Americas [10]. The United States and Canada were included as high-income comparator countries. Data from 2019 were used exclusively, as this was the most recent year with comprehensive coverage across all indicators.

The countries included in this analysis are part of the Non-Latin Caribbean region, comprising primarily English- and French-speaking small island developing states and coastal nations. Although geographically spread across the Caribbean Sea and Atlantic Ocean, these countries share common historical, cultural, and economic ties, including membership in regional organizations such as the Caribbean Community (CARICOM). Most have majority-Black populations and face similar structural challenges, including limited healthcare infrastructure, small population sizes, and a growing burden of chronic disease. Grouping these nations together allows for meaningful regional comparisons and helps highlight broader public health trends across this understudied region.

The dataset included mortality measures for major CVD conditions, such as ischemic heart disease, stroke, hypertensive heart disease, and cardiomyopathy, as well as select kidney diseases (S1 Table). Indicators were disaggregated by sex and included deaths and years of life lost (YLLs). Behavioral and clinical risk factor data were also extracted, including smoking rates, alcohol consumption, lipid profiles, physical activity, and hypertension (HTN) prevalence, diagnosis, treatment, and control.

These data were selected for their consistency, regional comparability, and relevance to health disparities research in small island nations, where individual-level datasets are often limited or unavailable. Although the original PAHO ENLACE dataset includes lower and upper uncertainty bounds for many indicators, these were not retained for analysis and are not reported in this study. PAHO does not specify whether these bounds represent formal confidence intervals, but they reflect variability in the underlying data and estimation process. Additionally, we analyzed national-level NCD prevention policy implementation using the 2022 PAHO ENLACE interactive scorecard. The scorecard tracks progress on commitments from the 2011 UN Political Declaration and the 2014 UN Outcome Document on NCDs. We focused on the following indicators: (1) CVD drug therapy (availability and accessibility of essential cardiovascular medications), (2) trans-fat policies (regulations on trans fats in food products), (3) salt policies (measures to reduce salt intake, such as sodium reduction standards or mandatory labeling), (4) alcohol advertising restrictions (policies limiting alcohol advertising, especially to minors), (5) alcohol availability (restrictions on alcohol sale and distribution), (6) tobacco advertising bans (prohibitions on tobacco advertising), (7) tobacco health warnings (requirements for graphic or textual warnings on tobacco packaging), (8) NCD policy plans (government strategies to combat NCDs), (9) surveillance (systems for monitoring NCD risk factors and outcomes), and (10) mortality (national statistics on NCD-related deaths). Each indicator was classified by PAHO as fully implemented (green), partially implemented (yellow), not implemented (red), or unknown (grey).

## Data analysis

All data cleaning, visualization, and statistical analyses were conducted using Microsoft Excel and Python. Descriptive statistics were used to summarize health outcome and risk factor distributions across countries, disaggregated by sex where available.

A geographic map was created in QGIS 3.34 T using GADM shapefiles to visualize CVD mortality rates across subregions performed by Fiver professional Ayesha Suraweera. No statistical analyses were performed in QGIS; the map presents descriptive values extracted from PAHO [11,12].

To assess sex-based differences within the Caribbean (CAR), two-tailed Student's t-tests were performed to compare male and female values across multiple health indicators, including mortality and clinical risk factors. For regional comparisons, Welch's t-tests were used to test differences between CAR countries and North America (NA), defined here as the United States and Canada. Statistical significance was defined as $p < 0.05$ [13].

## Results

### The Caribbean has elevated CVD mortality compared to North America

To understand how CVD outcomes differ by region, we began by mapping age-standardized CVD mortality rates across the Americas (Fig 1). The Non-Latin Caribbean had the highest mortality rate (196.7 per 100,000), higher than rates in North America (122.6), the Andean Area (128.9), the Southern Cone and Brazil (151.4), and Central America, Mexico, and the Latin Caribbean (169.6).

   Building on this regional context, we then examined sex-disaggregated data on hemorrhagic and ischemic stroke disease mortality between the CAR and NA (Fig 2A–2B). Hemorrhagic stroke mortality in men was nearly five times higher in CAR than in NA (46.6 vs. 10.0; p < .001), with similarly elevated rates among women (37.1 vs. 8.0; p < .01). Ischemic stroke mortality was also significantly higher in CAR for both sexes (p < .001). Hypertensive heart disease showed notable

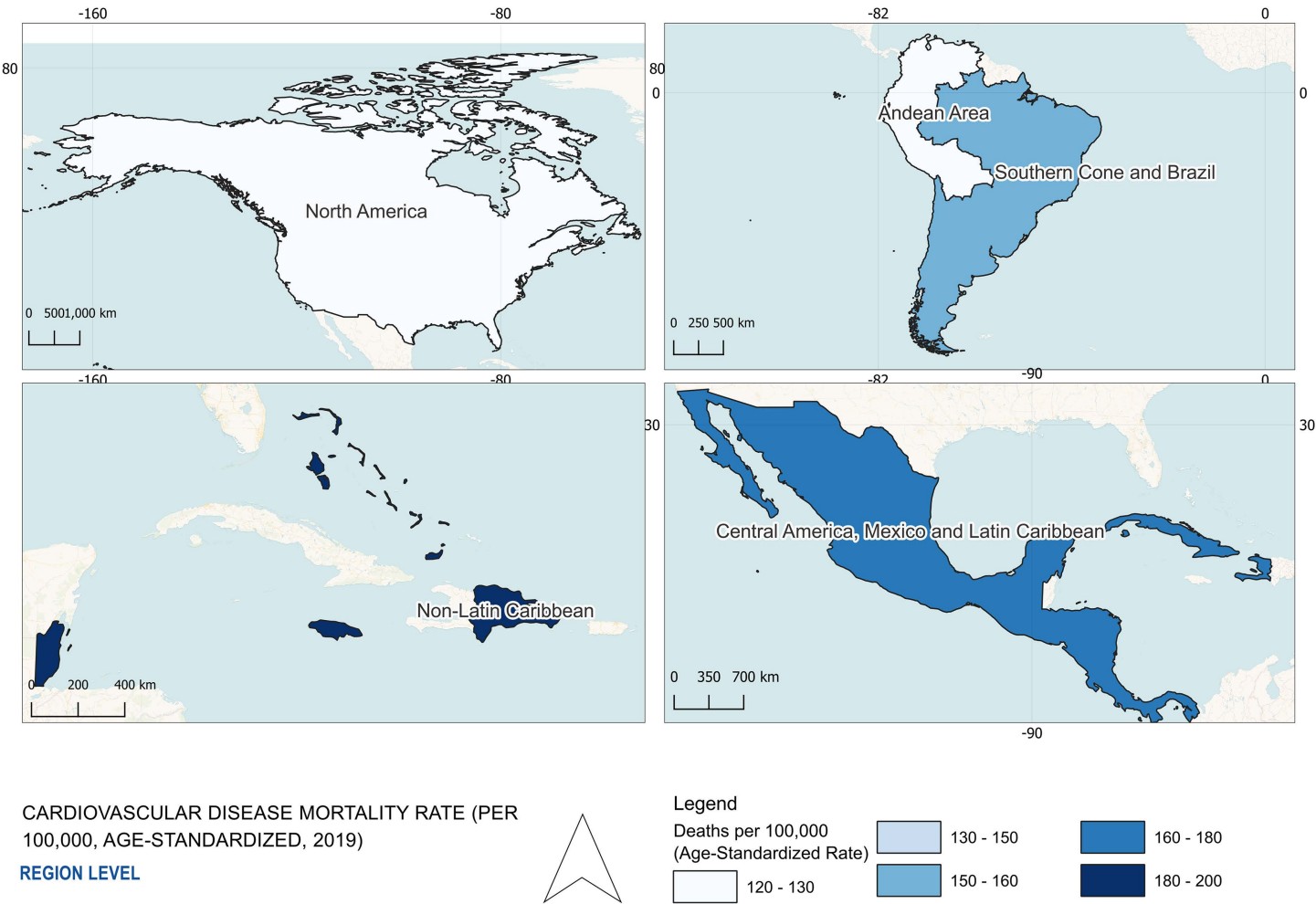

**Fig 1. Cardiovascular Disease Death Rates by AMRO Subregion (2019) All data were obtained from the PAHO ENLACE dataset, reporting cardiovascular disease (CVD) mortality rates per 100,000 population for the year 2019.** The dataset includes age-standardized death rates for CVD's across distinct regions in the Americas. Darker shades indicate higher CVD mortality, highlighting significant regional disparities. The map was created in QGIS 3.34 T using shapefiles from GADM. The Non-Latin Caribbean region demonstrates the highest CVD burden relative to other American subregions.

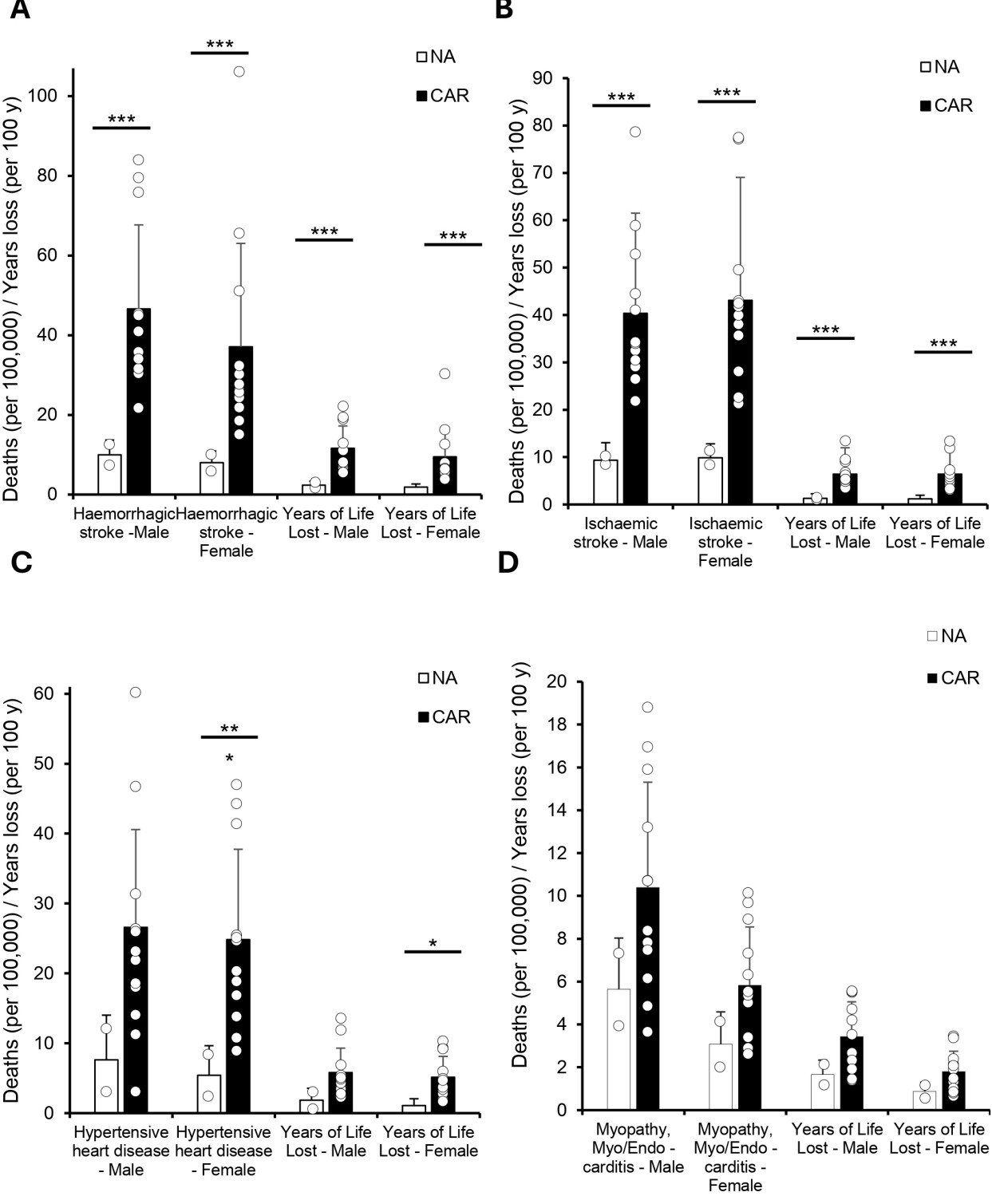

**Fig 2. The Caribbean Faces a Higher CVD Mortality Burden** All data were obtained from the 2019 PAHO ENLACE dataset, which compiles national health metrics on CVD mortality and years of life lost (YLL) due to premature death. CVD deaths are reported per 100,000 people, while YLL is expressed per 100 years. A Welch's t-test was used to compare North America (NA: United States and Canada) and the Caribbean (CAR: Antigua and Barbuda, the Bahamas, Barbados, Belize, Dominica, Grenada, Guyana, Haiti, Jamaica, Saint Kitts and Nevis, Saint Lucia, Saint Vincent and the

Grenadines, Suriname, and Trinidad and Tobago) due to unequal variance between groups. Statistical significance is denoted as p<0.05 (), p<0.01 (), and p<0.001 (). **(A)** Bar graph comparing mortality from hemorrhagic stroke and ischemic stroke between NA and CAR, showing significantly higher mortality in the Caribbean for both stroke types in men and women (p<0.001 for all). **(B)** Years of life lost due to hemorrhagic and ischemic stroke, which was significantly higher in the Caribbean across both sexes (p<0.001 for all). **(C)** Mortality due to hypertensive heart disease was significantly higher in Caribbean men and women (p<0.001 for men, p<0.05 for women), while YLL was significantly higher in women (p<0.05). **(D)** Mortality from cardiomyopathy, myocarditis, and endocarditis showed no significant differences between NA and CAR.

disparities as well (Fig 2C). Mortality was over three times higher in CAR than NA for women (24.8 vs. 5.4; p<.01), with a similar trend in men that was not significant (26.6 vs. 7.6; p=0.051).

Next, we analyzed the rates of cardiomyopathy, myocarditis, and endocarditis (Fig 2D). While mortality and years of life lost were consistently higher in the Caribbean, none of these differences reached statistical significance.

## Behavioral risk profiles differ by gender in the Caribbean and do not fully explain higher CVD mortality

To further explore factors influencing regional CVD mortality disparities, we assessed differences in behavioral and lifestyle risk factors between the CAR and NA such as alcohol consumption per capita and Alcohol-related deaths (Fig 3A). Alcohol consumption among men was significantly lower in CAR compared to NA (11.0 vs. 14.6; p<.05), although alcohol-related mortality was notably higher (17.0 vs. 11.4; p<.05). Female alcohol consumption showed a similar but non-significant trend in the CAR (2.9 vs. 4.4; p=.07).

We next examined metabolic risk factors for CVD (Fig 3B). Physical inactivity rates did not differ significantly between regions for either sex, with Caribbean women reporting 43.0% compared to 38.1% in North America (p=.06). Overweight prevalence among men was lower in CAR than NA (51.4% vs. 69.7%; p=.09), and female overweight prevalence was slightly higher in CAR, though this difference was not statistically significant. In evaluating substance use, opioid and cocaine use rates were consistently lower in the Caribbean compared to North America, though these differences were not statistically significant (Fig 3C).

When comparing lifestyle behaviors between sexes within the Caribbean, significant differences emerged (Fig 3D). Tobacco use was substantially higher among men than women (14.6% vs. 2.4%; p<.01). Men also exhibited significantly higher alcohol consumption (11.0 vs. 2.9; p<.001), alcohol-related deaths (17.0 vs. 4.2; p<.001), and drug use rates (1.6% vs. 0.4%; p<.001). In contrast, women had a significantly higher prevalence of overweight compared to men (43.2% vs. 34.0%; p<.001).

## Cholesterol risk profiles are less favorable in the Caribbean despite lower trans fat and salt intake

Given the strong links between cholesterol levels and CVD outcomes [14], we analyzed cholesterol profiles and dietary patterns across regions and sexes. The Caribbean (CAR) exhibited significantly higher Total-to-HDL cholesterol ratios compared to North America (NA) for both men (4.0 vs. 3.6; p<.001) and women (3.8 vs. 3.0; p<.001). Mean non-HDL cholesterol was also significantly higher among Caribbean women (3.6 vs. 3.2; p<.01), with a similar but non-significant trend among men (3.4 vs. 3.3).

Sex-specific cholesterol comparisons within the Caribbean revealed notable differences. Women had significantly higher mean HDL cholesterol levels (1.23 vs. 1.11; p<.001) and total cholesterol (4.7 vs. 4.4; p<.001) than men. In contrast, men had a significantly higher total-to-HDL cholesterol ratio (4.0 vs. 3.8; p<.05). Mean non-HDL cholesterol was also significantly higher in Caribbean women compared to men (3.6 vs. 3.4; p<.05).

Exploring dietary patterns, we observed that trans-fatty acid intake was lowest in the Caribbean compared to other American regions, with minimal differences between sexes. Total dietary energy from fat was also lower in the Caribbean than in North America, particularly among men (6.4% vs. 9.0%) and women (8.0% vs. 11.2%).

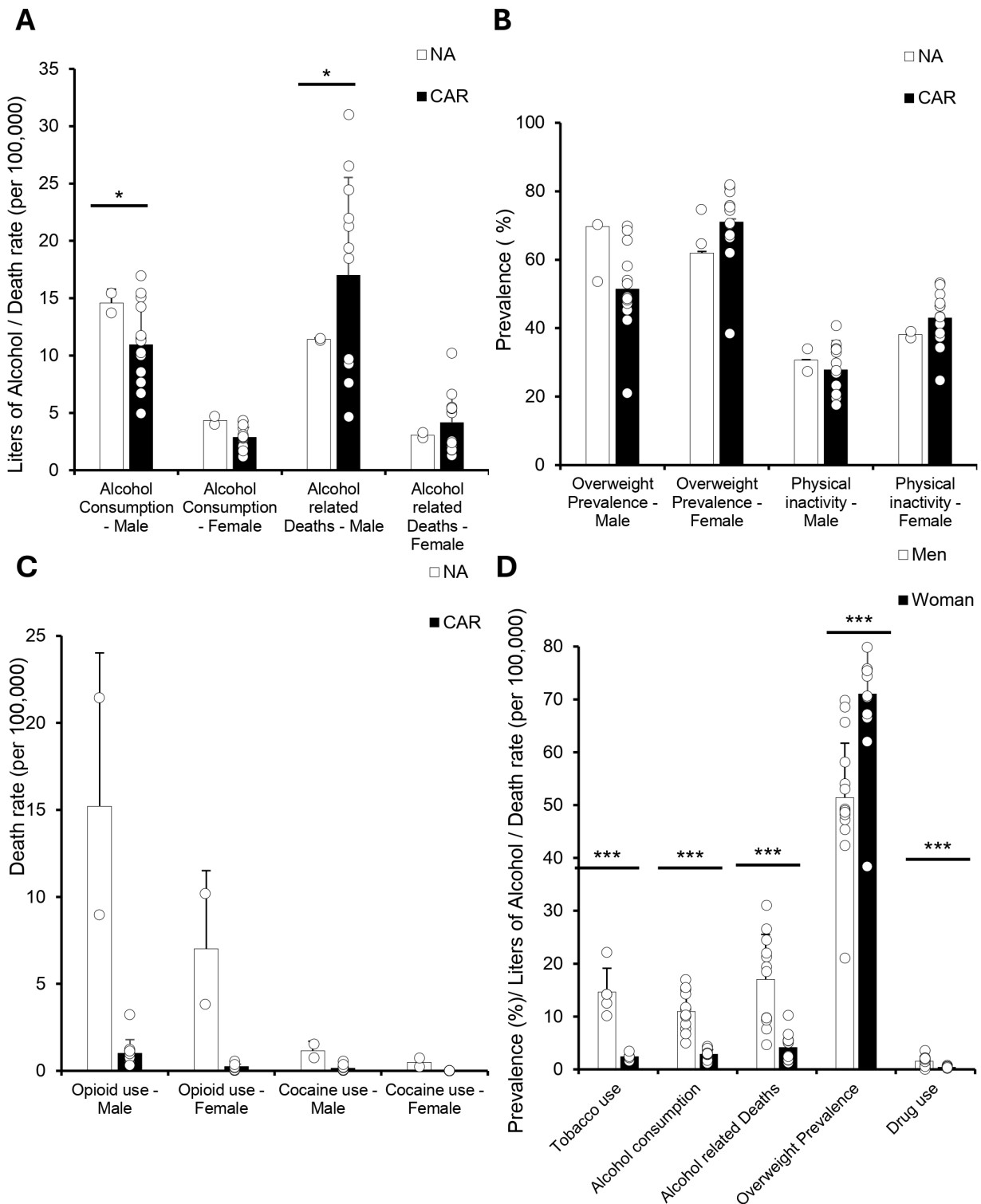

**Fig 3. Caribbean Men Have Higher Substance Use, While Women Show Greater Overweight Prevalence** All data were obtained from the 2019 PAHO ENLACE dataset, which reports national health metrics on alcohol consumption, overweight prevalence, physical inactivity, and drug use, including opioids and cocaine. **(A)** Bar graph of age-standardized alcohol consumption per capita (liters) for individuals 15 years and older, alongside alcohol-related deaths per 100,000. Alcohol consumption and alcohol-related deaths were higher in Caribbean men compared to North

American men ($p<0.05$, $p<0.05$), with no significant differences in women. **(B)** Bar graph of age-standardized overweight prevalence (BMI ≥ 25) and self-reported physical inactivity. We did not observe any significant changes. **(C)** Bar graph comparing opioid and cocaine use between North America and the Caribbean, showing no significant differences due to high variance. **(D)** When comparing Caribbean men against women, a Student's t-test showed men had higher rates of cigarette/tobacco smoking, alcohol consumption, alcohol-related deaths, and overall drug use ($p<0.001$ for all).Women had a significantly higher prevalence of overweight ($p<0.001$). Physical inactivity was excluded from sex-based comparisons due to a lack of statistical significance.

### HTN control remains poor and kidney disease mortality elevated in the Caribbean compared to North America

To better understand the management of CVD risk factors, we first examined HTN prevalence and control between the CAR and NA (Fig 5A–5B). Undiagnosed HTN prevalence was significantly higher in CAR men compared to NA men (47.6% vs. 20.8%; p<.001), while women showed a smaller, non-significant difference. Controlled HTN prevalence was notably lower in CAR compared to NA for both sexes, with significant differences observed among women (25.3% vs. 54.0%; p<.001) and a substantial but non-significant difference among men (15.8% vs. 54.4%). Similarly, HTN treatment rates were lower in CAR for both sexes compared to NA, achieving statistical significance among women (57.4% vs. 72.0%; p<.01) and demonstrating a large but non-significant gap among men (37.8% vs. 71.9%).

We also assessed mortality due to chronic kidney disease (CKD) related to diabetes and overall kidney disease (Fig 5C–5D). CKD due to diabetes mortality was higher in the Caribbean compared to North America for both men (12.8 vs. 3.2 per 100,000) and women (9.8 vs. 2.4 per 100,000), though these differences were not statistically significant. In contrast, mortality from overall kidney disease was significantly higher in Caribbean men (35.5 vs. 10.6 per 100,000; p<.05) and women (25.1 vs. 7.8 per 100,000; p<.05).

### Healthcare system indicators reveal major gaps in NCD management across the Caribbean

To contextualize CVD disparities, we assessed regional progress in implementing key healthcare indicators related to noncommunicable diseases (NCDs). North America demonstrated strong performance, fully achieving implementation for nearly all evaluated indicators, including NCD mortality tracking, surveillance systems, and national policy frameworks for CVD prevention and treatment (Fig 6A).

In contrast, Caribbean countries showed substantial systemic gaps across nearly all evaluated indicators (Fig 6B). Most Caribbean countries had limited or no implementation of critical healthcare system measures, including tobacco advertising bans (10 countries with no implementation), alcohol advertising restrictions (13 countries with no implementation), salt reduction policies (11 countries with no implementation), and trans-fat policies (11 countries with no implementation). Similarly, guidelines for NCD management and CVD drug therapy were either partially implemented or completely lacking across the region.

## Discussion

This study revealed three key findings. First, cardiovascular mortality rates were consistently higher in the Non-Latin Caribbean compared to North America, with particularly elevated mortality from stroke and hypertensive heart disease, especially among women (Figs 1–2). Second, disparities in behavioral and clinical risk factor profiles were evident: Caribbean men had higher alcohol-related CVD mortality despite lower reported consumption, while Caribbean women exhibited higher prevalence of overweight, physical inactivity, and more atherogenic lipid profiles (Figs 3–5). Third, significant gaps in national-level NCD policy implementation were observed in the Caribbean, including limited adoption of WHO-recommended measures such as tobacco and alcohol regulations, salt and trans-fat reduction initiatives, and clinical guidelines for hypertension and CVD management (Fig 6). Together, these findings suggest that elevated CVD risk in the Caribbean may be shaped by a combination of sex-specific biological vulnerability, underdiagnosed clinical conditions, and structural gaps in both healthcare delivery and policy implementation.

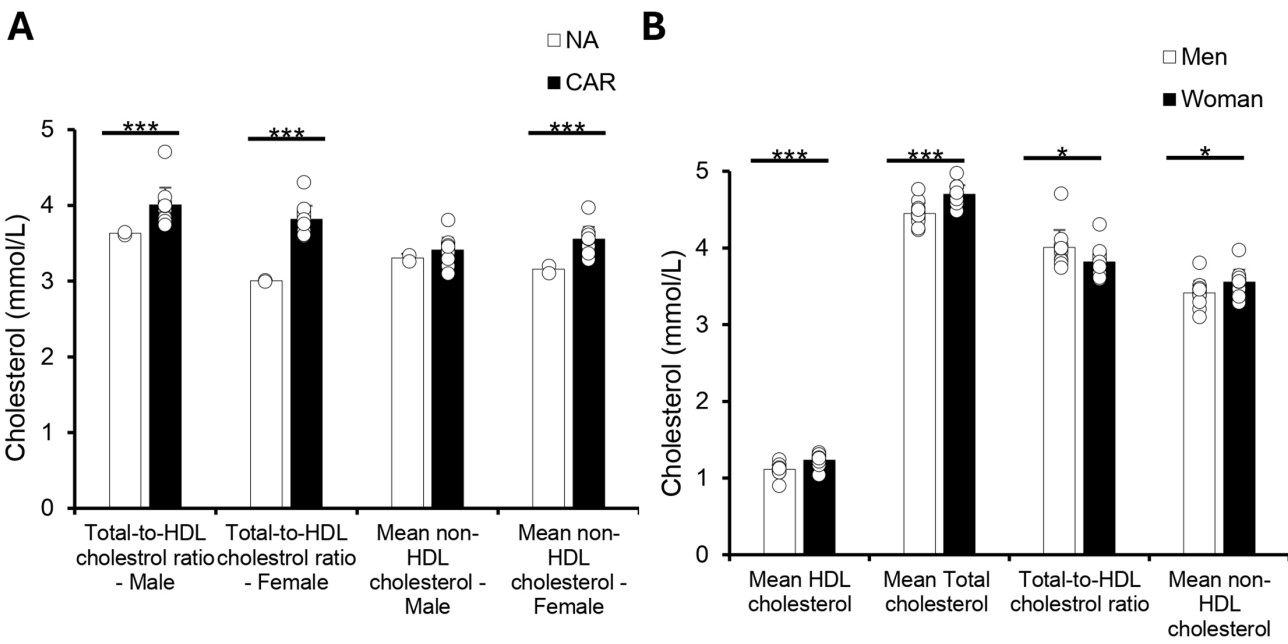

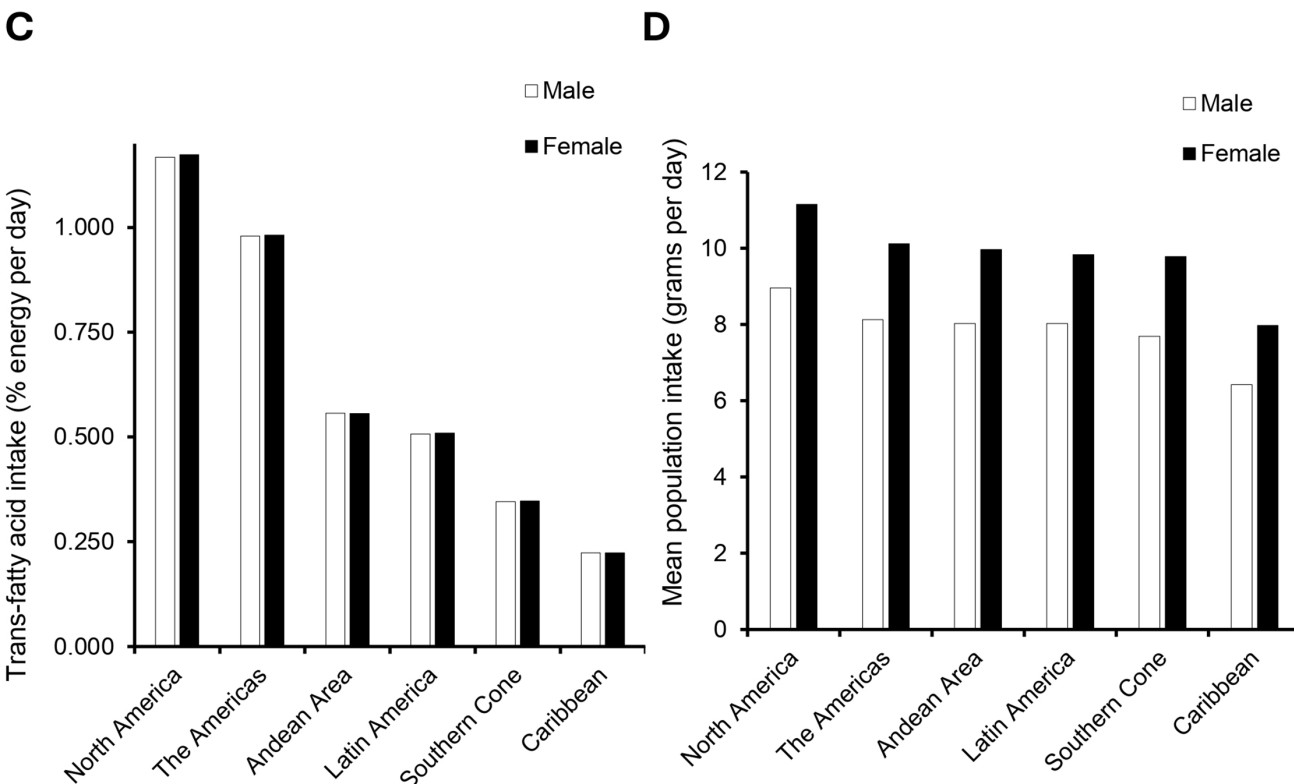

**Fig 4. Caribbean Adults Have Worse Cholesterol Profiles but Lower Trans-Fat and Dietary Intake Than Other Regions** All data were obtained from the 2019 PAHO ENLACE dataset, which reports national health metrics on cholesterol levels, trans-fatty acid intake, and mean population dietary intake. **(A)** Bar graph comparing total-to-HDL cholesterol ratio and mean non-HDL cholesterol between NA and CAR. Total-to-HDL cholesterol ratio was significantly higher in Caribbean men ($p = 0.0002$) and women ($p = 0.0003$) compared to North American men and women. Mean non-HDL

cholesterol was significantly higher in Caribbean women compared to North American women ($p = 0.0004$), but no significant difference was observed in men. **(B)** Bar graph comparing mean HDL cholesterol, total cholesterol, total-to-HDL cholesterol ratio, and mean non-HDL cholesterol between men and women in the Caribbean. Mean HDL ($p = 0.0001$) and total cholesterol ($p = 0.0002$) were significantly higher in women, while total-to-HDL cholesterol ratio ($p = 0.03$) and mean non-HDL cholesterol ($p = 0.04$) were significantly higher in men. **(C)** Bar graph ranking trans-fatty acid intake (% energy per day) across regions, showing the highest intake in North America, followed by the Americas region, Andean Area, Latin America, Southern Cone, and the lowest in the Caribbean. **(D)** Bar graph ranking mean population dietary intake (grams per day) across regions, with the highest intake in the Southern Cone, followed by Latin America, the Andean Area, the Americas, North America, and the lowest intake in the Caribbean.

This study revealed significant disparities in CVD mortality across the Caribbean, particularly highlighting elevated rates of stroke and hypertensive heart disease compared to North America [1,2]. The observed higher mortality in Caribbean populations, including significantly elevated rates of hypertensive heart disease among women, emphasizes the urgent need to better understand and address the unique CVD risk profiles within this region [3,15]. Although CVD remains the leading cause of death across the Americas, the Caribbean has experienced stagnation or worsening in outcomes despite global advances in CVD prevention and care, as exemplified by substantial healthcare improvements in North America [6] and projected CVD burdens driven by demographic shifts such as aging populations and increasing obesity [7]. By examining sex-specific CVD mortality and risk factors in a regionally disaggregated manner, this study addresses critical research gaps and enhances understanding of how systemic inequities and biological factors intersect to shape CVD health disparities [5]. The subsequent discussion integrates these findings into the broader regional context, evaluates current literature, and identifies areas for targeted interventions and future research.

Our analysis confirmed markedly elevated CVD mortality in the Caribbean compared to North America, particularly for stroke and hypertensive heart disease. These findings align with earlier regional studies that reported consistently higher CVD burdens in Caribbean populations [1,4]. However, our detailed sex-specific breakdown highlights an additional layer of complexity not fully explored in prior research. For instance, Caribbean women exhibited significantly higher mortality from hypertensive heart disease compared to their North American counterparts, reinforcing concerns raised by previous studies regarding gender disparities in CVD outcomes [3,9]. Furthermore, the pronounced excess in hemorrhagic and ischemic stroke mortality in both sexes extends beyond existing evidence, which often aggregated Caribbean data within broader Latin American contexts, masking important subregional differences [4]. By disaggregating data specifically for the Non-Latin Caribbean, our results clarify the scale and scope of CVD disparities, underscoring the urgent need for targeted public health strategies tailored to this uniquely vulnerable region.

Our results illustrated important disparities in behavioral and metabolic risk factors between the Caribbean and North America, particularly highlighting mismatches between risk exposures and CVD outcomes. Notably, Caribbean men reported significantly lower alcohol consumption yet experienced higher alcohol-related CVD mortality compared to North American men, suggesting complexities beyond mere consumption volume, such as drinking patterns or healthcare access influencing outcomes [16]. This finding complements earlier global analyses indicating that lifestyle factors alone may not fully account for regional CVD disparities, especially in low-resource settings [17,18]. Similarly, Caribbean women exhibited higher prevalence of overweight and physical inactivity, patterns consistent with prior Caribbean-focused literature identifying excess body weight as a persistent contributor to CVD morbidity [3,5]. However, despite similar or even lower prevalence of certain behavioral risk factors compared to North America, the Caribbean faces notably worse CVD mortality. This paradox echoes findings from LMICs showing that younger age at stroke is strongly associated with healthcare access and quality, highlighting the role of systemic barriers in shaping cardiovascular risk profiles [19]. Thus, our study underscores the need for deeper investigations into healthcare access and system-level interventions in addition to traditional risk-factor modification.

Our findings indicated significant regional differences in cholesterol profiles, HTN management, and kidney disease mortality, which may help explain some of the observed CVD disparities. Elevated total-to-HDL cholesterol ratios and

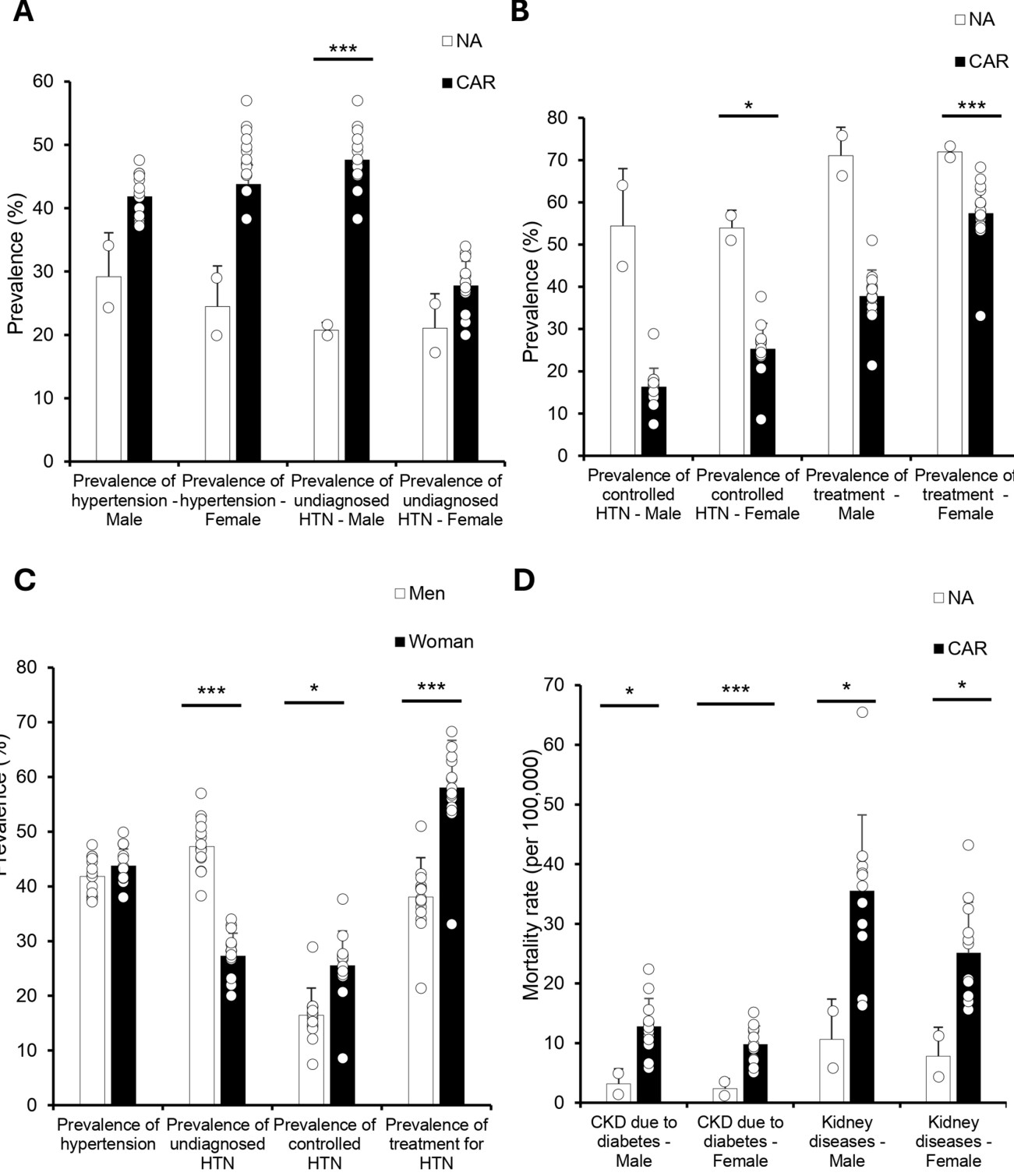

**Fig 5. Undiagnosed HTN is Higher in Caribbean Men, While CKD Mortality is Elevated in Both Sexes All data were obtained from the 2019 PAHO ENLACE dataset, which reports national health metrics on hypertension (HTN), treatment, and chronic kidney disease (CKD) mortality. (A)** HTN prevalence showed no significant differences between NA and CAR, but undiagnosed HTN was significantly higher in Caribbean men compared to North American men ($p < 0.001$), with no significant difference in women. **(B)** Controlled HTN was significantly lower in Caribbean women

compared to North American women ($p < 0.05$), and treatment prevalence was also significantly lower in Caribbean women ($p < 0.001$). Caribbean men trended toward lower treatment prevalence but did not reach statistical significance ($p = 0.06$). **(C)** No significant differences were observed in HTN prevalence between men and women, but undiagnosed HTN was significantly higher in men ($p < 0.001$). Women had significantly higher rates of controlled HTN ($p < 0.05$) and HTN treatment ($p < 0.001$). **(D)** CKD mortality due to diabetes was significantly higher in CAR for both men ($p < 0.05$) and women ($p < 0.001$), and overall kidney disease mortality was also significantly higher in both sexes ($p < 0.05$ for men, $p < 0.05$ for women).

increased non-HDL cholesterol levels, particularly among Caribbean women, represent biologically plausible mechanisms contributing to higher cardiovascular risk in this population. Although prior global studies have broadly associated dyslipidemia with increased cardiovascular risk [8,9], few have specifically examined lipid patterns in Caribbean populations, leaving an important knowledge gap regarding regional variations in cholesterol metabolism or genetic predispositions. A systematic review encompassing Latin America and the Caribbean reported a high prevalence of low HDL-cholesterol at 48%, underscoring the need for targeted research in specific subregions, including the Caribbean [20].

Additionally, our results demonstrated major gaps in HTN management in the Caribbean, with significantly higher rates of undiagnosed hypertension among men and significantly lower control and treatment rates among women. These findings align with previous literature highlighting systemic inadequacies in HTN screening, diagnosis, and control within Caribbean healthcare systems [3,21]. Furthermore, mortality from chronic kidney disease due to diabetes was higher in Caribbean populations, though not statistically significant, while overall CKD mortality was significantly elevated in both men and women. Multiple studies have identified diabetes and hypertension as key drivers of CKD burden in the region [22,23], while recent evidence also implicates APOL1 risk alleles and environmental stressors in disease progression [24]. Despite the regional importance of CKD, population-level data remain sparse, with most studies focused on single countries or small cohorts. By providing standardized mortality comparisons across 15 Caribbean nations, this study helps fill a critical gap in regional kidney health surveillance. These findings highlight the need for more integrated strategies to address both cardiovascular and renal complications. Taken together, these results emphasize that biological risk factors, compounded by systemic healthcare deficiencies, may play a critical role in driving CVD health disparities in this region, highlighting the urgent need for integrated interventions addressing both biological and systemic determinants of health.

This interplay of biological risk and systemic gaps is further compounded by deeply rooted healthcare access barriers that limit diagnosis, management, and continuity of care across the Caribbean. In Haiti, individuals with heart failure face systemic challenges—including limited transportation, cost burdens, and clinic shortages—that hinder long-term disease control [25]. Across the broader region, disparities in noncommunicable disease outcomes are strongly associated with fragmented health systems and uneven service distribution [26], while socioeconomic vulnerability among Caribbean women further restricts engagement with care and chronic disease prevention [27,28]. These patterns reflect longstanding regional gaps in structural health equity, particularly for women and low-income populations. Our prior work on diabetes and obesity in the Caribbean showed that undiagnosed diabetes was significantly more prevalent in countries with lower GDP per capita, with The Bahamas reporting among the highest rates despite relatively strong national income. We also found that food insecurity, income inequality, and high out-of-pocket costs were strongly correlated with diabetes-related mortality. In that study, Caribbean women had markedly higher overweight prevalence than men, suggesting that gendered socioeconomic barriers may contribute to chronic disease disparities. While our current findings did not show women to be disproportionately affected across all outcomes, these socioeconomic dynamics likely compound biological and systemic risks in complex ways.

Our findings highlight critical deficiencies in healthcare infrastructure, including inadequate detection and control of hypertension and insufficient chronic disease management systems. These limitations align closely with the observed regional deficits in implementing critical noncommunicable disease (NCD) management indicators, such as tobacco advertising bans, alcohol advertising restrictions, salt reduction initiatives, and trans-fat policies [3,6].

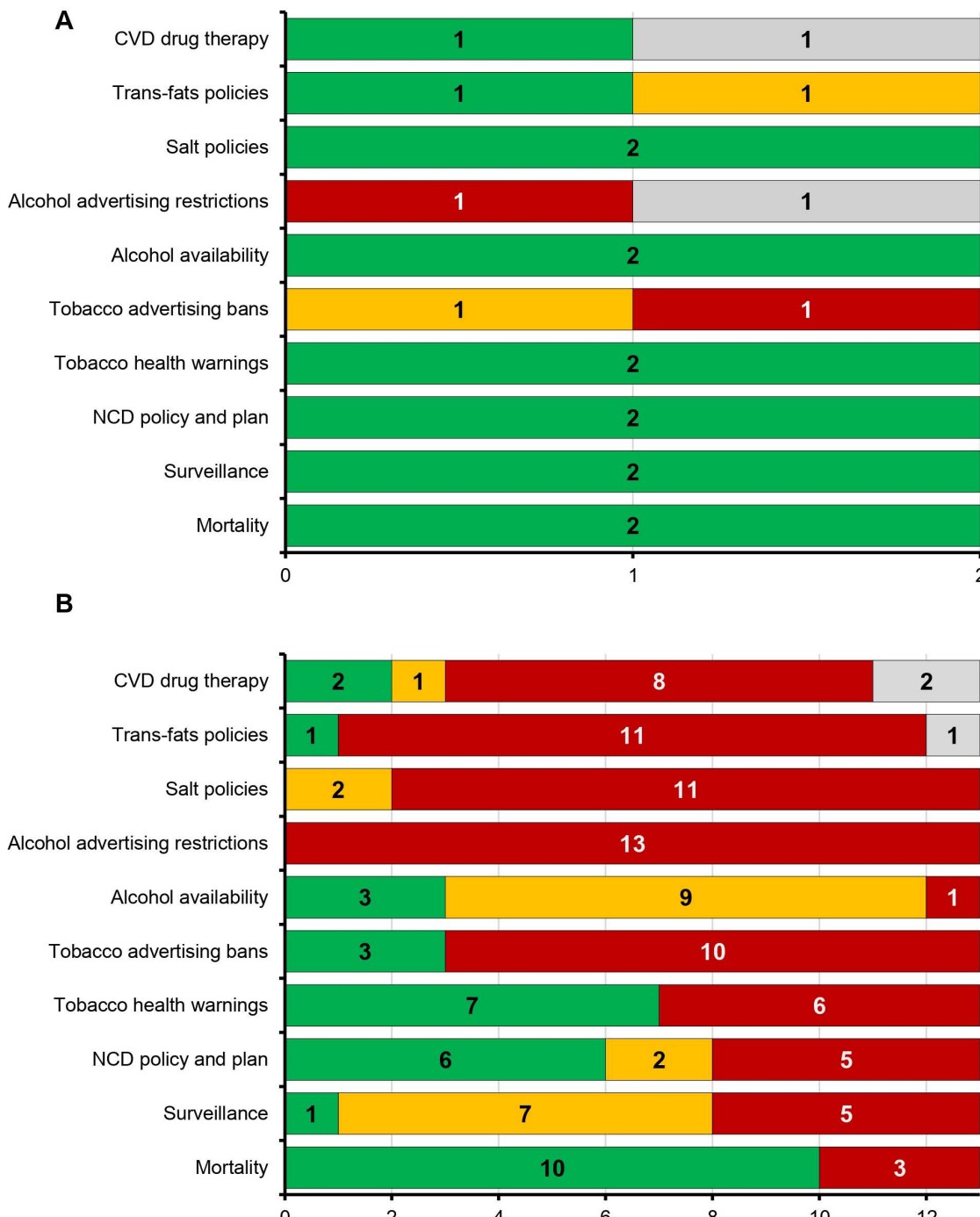

**Fig 6. Caribbean countries lag behind North America in implementation of key NCD progress indicators** All data were sourced from the PAHO ENLACE 2022 dataset, summarizing regional progress on noncommunicable disease (NCD) indicators. The indicators include healthcare system measures and clinical guidelines related specifically to CVD and general NCD management. Green indicates "fully achieved," yellow indicates "partially achieved," red indicates "not achieved," and gray represents missing data. **(A)** North America shows strong implementation of NCD progress indicators, with most fully achieved. **(B)** The Caribbean demonstrates significantly poorer implementation across nearly all indicators, highlighting systemic gaps in healthcare infrastructure, clinical practice guidelines, primary healthcare funding, risk stratification, and patient education. The marked regional differences underscore potential contributors to CVD disparities observed between these regions.

PLOS Global Public Health

Enhancing primary care accessibility, particularly in underserved and rural areas, and investing in affordable, scalable screening methods like home blood pressure monitoring could significantly bolster early detection and effective management of CVD risks. Additionally, the development of region-specific clinical guidelines incorporating sex-specific and culturally sensitive CVD prevention strategies is essential to address unique risk profiles and biological susceptibilities characteristic of Caribbean populations [8,9,15].

Given the elevated rates of undiagnosed hypertension and significantly higher kidney disease mortality observed in this study, implementing integrated care models that concurrently address CVD, diabetes, and renal complications could prove more effective than current fragmented approaches [17,19]. Ultimately, comprehensive policy reform prioritizing equitable healthcare access [29], robust surveillance systems, targeted public education initiatives, and biologically informed preventive measures represents the most promising pathway to reducing CVD disparities and improving population health outcomes across the Caribbean region.

This study has several limitations that should be considered when interpreting the findings. First, its cross-sectional design restricts causal inference, allowing only for associations rather than direct cause-and-effect conclusions. Additionally, reliance on publicly available aggregated data from international sources limits the granularity and potential accuracy of region-specific estimates, particularly in smaller Caribbean nations with inconsistent data collection infrastructure [5]. This was especially true for Dominica and Saint Kitts and Nevis, where limited data availability made it difficult to compare them across all indicators. Furthermore, due to limited regional data availability, important socioeconomic determinants such as healthcare quality, access disparities, and cultural factors influencing health behaviors could not be comprehensively analyzed. Additionally, despite diabetes being a major contributor to cardiovascular morbidity, no cardiovascular-specific diabetes prevalence metric was available in the PAHO ENLACE dataset [30].

Future research should address these gaps by conducting longitudinal, prospective studies within Caribbean contexts, incorporating more detailed socioeconomic, genetic, and health system-level data to clarify mechanisms underlying observed disparities [18,21]. Moreover, given the distinct sex-based differences highlighted in our findings, there is a need for more sex-disaggregated research, including targeted clinical trials and qualitative studies exploring gender-specific barriers to care, symptom presentation, and treatment responsiveness within Caribbean populations. Expanding such research would significantly strengthen the evidence base necessary for creating targeted and effective CVD health policies in the region.

## Supporting information

**S1 Table. Country-level burden of cardiovascular and kidney diseases and associated risk factors in the Caribbean and North America All data were sourced from the PAHO ENLACE 2019 dataset, summarizing national mortality, disability-adjusted life years (DALYs), years of life lost (YLLs), and years lived with disability (YLDs) for major cardiovascular and kidney diseases, stratified by sex.** Indicators also include behavioral risk factors (tobacco use, alcohol consumption, physical inactivity, overweight/obesity, cholesterol) and hypertension awareness, treatment, and control estimates. This table provides the quantitative foundation for regional comparisons of cardiometabolic disease burden and risk profiles across Caribbean and North American countries.
(XLSX)

## Author contributions

**Conceptualization:** Cesar Barrabi.

**Data curation:** Cesar Barrabi.

**Formal analysis:** Cesar Barrabi.

**Investigation:** Cesar Barrabi, Celia Foster.

**Methodology:** Cesar Barrabi.

**Project administration:** Cesar Barrabi.

**Resources:** Cesar Barrabi.

**Software:** Cesar Barrabi.

**Supervision:** Cesar Barrabi.

**Validation:** Celia Foster.

**Visualization:** Cesar Barrabi.

**Writing – original draft:** Cesar Barrabi.

**Writing – review & editing:** Cesar Barrabi, Celia Foster.

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
