## [Decision Letter · Decision Letter 0]

24 Jun 2025

PGPH-D-25-00790

Elevated Cardiovascular Risk Factors and Chronic Disease Mortality in the Caribbean: A Cross-Sectional Study

Dear Dr. BARRABI,

Thank you for submitting your manuscript to PLOS Global Public Health. After careful consideration, we feel that it has merit but does not fully meet PLOS Global Public Health’s publication criteria as it currently stands. Therefore, we invite you to submit a revised version of the manuscript that addresses the points raised during the review process.

As one of the reviewers has suggested (see below), I think it would be really helpful if you clearly stated what the overall aim/objectives of this study are as this will make the study objectives very clear. Also, clearly address issues related to the differences in the imputed dataset and the complete dataset in terms of CVD and stroke rates and provide a succinct summary of the key findings of the manuscript in the first paragraph of the discussion section to enhance readability / understanding of the study’s key findings.

We look forward to receiving your revised manuscript.

Kind regards,

Ikechi G Okpechi

Academic Editor

Journal Requirements:

1. We noticed that you used “data not shown” in the manuscript. We do not allow these references, as the PLOS data access policy requires that all data be either published with the manuscript or made available in a publicly accessible database. Please amend the supplementary material to include the referenced data or remove the references.

Reviewers' comments:

Reviewer's Responses to Questions

**Comments to the Author**

1. Does this manuscript meet PLOS Global Public Health’s publication criteria? Is the manuscript technically sound, and do the data support the conclusions? The manuscript must describe methodologically and ethically rigorous research with conclusions that are appropriately drawn based on the data presented.

Reviewer #1: Yes

Reviewer #2: Yes

2. Has the statistical analysis been performed appropriately and rigorously?

Reviewer #1: N/A

Reviewer #2: Yes

3. Have the authors made all data underlying the findings in their manuscript fully available (please refer to the Data Availability Statement at the start of the manuscript PDF file)?

Reviewer #1: Yes

Reviewer #2: Yes

4. Is the manuscript presented in an intelligible fashion and written in standard English?

Reviewer #1: Yes

Reviewer #2: Yes

5. Review Comments to the Author

Reviewer #1: Barrabi et al, performed a cross-sectional analysis from publicly available health data from PAHO to examine cardiovascular disease prevalence, mortality, behavioral and clinical risk factor burden, and identify sex differences in these outcomes and examine national level policy implementation for NCD prevention across the Caribbean. While this data provides important data on a region which is often understudied, a few comments/concerns are raised from review of the manuscript, itemized below:

Introduction

1. Introduction should exclude preliminary findings/results (lines 54-57), suggest ending the introduction with the study objectives/research question and a brief description of how question was answered/objectives met as a segway to the methods.

Methods

2. What were the overall aim/objectives of this study? Not explicitly stated and while I gleaned from review of the manuscript it seems to be CVD mortality and risk factors, (see my opening paragraph). It may be useful to explicitly state this to enhance the readability. This is also in keeping with STROBE guidelines for reporting of the observational studies.

3. Diabetes mellitus is a major driver of cardiovascular disease burden worldwide, with rates increasing in the Caribbean, and an estimated 40% with unknown or untreated diabetes. (1,2) Given this, on what basis did we exclude this from the clinical risk factors, whilst including hypertension, dyslipidemia? Could increasing rates of untreated diabetes also add to the increasing CVD mortality?

4. Statistical methods:

Multiple imputation: The missing data seemed to be mainly from three countries. Were there differences in the imputed dataset and the complete dataset in terms of CVD and stroke rates? Secondly was a sensitivity analysis or any post analysis (e.g. Standardized differences) after imputation performed to ensure validity of the imputed dataset and ensure assumption met for imputation (missing at random)? This could be included a supplemental table.

Results

5. Would remove your interpretation of the results, or recommendations from the results section (lines 131-133, 142-143, 156-157, 161-162,174-175, 193-196) and keep this solely in the discussion.

6. Would dissuade use of the term “approached statistical significance” in your reporting. What were the measures of uncertainty for your estimates for age-standardized prevalence rates ?(eg 95% CI), could this be also reported in a main or supplemental table and in the main text.

Discussion

7. Suggest a summary of the key findings of the manuscript in the first paragraph, this will enhance readability of the manuscript.

8. See point 3 above, suspect diabetes burden adds to the attributable CVD mortality

9. Would expound on the missingness of the data from the countries as one of your limitations.

References

1. Bennett NR, Francis DK, Ferguson TS, Hennis AJ, Wilks RJ, Harris EN, MacLeish MM, Sullivan LW; U.S. Caribbean Alliance for Health Disparities Research Group (USCAHDR). Disparities in diabetes mellitus among Caribbean populations: a scoping review. Int J Equity Health. 2015 Feb 25;14:23. doi: 10.1186/s12939-015-0149-z. PMID: 25889068; PMCID: PMC4347914.

2. https://www.paho.org/en/news/5-9-2023-new-paho-analysis-reveals-diabetes-increasing-all-countries-americas

Reviewer #2: Abstract is OK

Introduction

Pages 54-57 should be deleted and taken to the summary

Methodology

There is need to describe the countries the geography, population and some important health indices of the Caribbean countries.

6. PLOS authors have the option to publish the peer review history of their article (what does this mean?). If published, this will include your full peer review and any attached files.

**Do you want your identity to be public for this peer review?** For information about this choice, including consent withdrawal, please see our Privacy Policy.

Reviewer #1: No

Reviewer #2: No

---

## [Decision Letter · Decision Letter 1]

1 Sep 2025

Elevated Cardiovascular Risk Factors and Chronic Disease Mortality in the Caribbean: A Cross-Sectional Study

PGPH-D-25-00790R1

Dear Dr. BARRABI,

We are pleased to inform you that your manuscript 'Elevated Cardiovascular Risk Factors and Chronic Disease Mortality in the Caribbean: A Cross-Sectional Study' has been provisionally accepted for publication in PLOS Global Public Health.

Best regards,

Ikechi G Okpechi

Academic Editor

Reviewer #3:

Reviewer Comments (if any, and for reference):

Reviewer's Responses to Questions

**Comments to the Author**

1. If the authors have adequately addressed your comments raised in a previous round of review and you feel that this manuscript is now acceptable for publication, you may indicate that here to bypass the “Comments to the Author” section, enter your conflict of interest statement in the “Confidential to Editor” section, and submit your "Accept" recommendation.

Reviewer #3: All comments have been addressed

2. Does this manuscript meet PLOS Global Public Health’s publication criteria? Is the manuscript technically sound, and do the data support the conclusions? The manuscript must describe methodologically and ethically rigorous research with conclusions that are appropriately drawn based on the data presented.

Reviewer #3: Yes

3. Has the statistical analysis been performed appropriately and rigorously?

Reviewer #3: Yes

4. Have the authors made all data underlying the findings in their manuscript fully available (please refer to the Data Availability Statement at the start of the manuscript PDF file)?

Reviewer #3: Yes

5. Is the manuscript presented in an intelligible fashion and written in standard English?

Reviewer #3: Yes

6. Review Comments to the Author

Reviewer #3: Data and analysis is informative and hopefully will inform policy changes at national level. well done to authors for doing this piece of research

7. PLOS authors have the option to publish the peer review history of their article (what does this mean?). If published, this will include your full peer review and any attached files.

**Do you want your identity to be public for this peer review?** For information about this choice, including consent withdrawal, please see our Privacy Policy.

Reviewer #3: No
